# Clinical Trials of Limbal Stem Cell Deficiency Treated with Oral Mucosal Epithelial Cells

**DOI:** 10.3390/ijms21020411

**Published:** 2020-01-09

**Authors:** Joan Oliva, Fawzia Bardag-Gorce, Yutaka Niihara

**Affiliations:** 1Emmaus Life Sciences, Inc., 21250 Hawthorne Blvd. Suite 800, Torrance, CA 90503, USA; yniihara@emmauslifesciences.com; 2Department of Medicine, Lundquist Institute, Torrance, CA 90502, USA; fgorce@lundquist.org

**Keywords:** cell sheet, limbal stem cell deficiency, tissue engineering, clinical trial

## Abstract

The corneal surface is an essential organ necessary for vision, and its clarity must be maintained. The corneal epithelium is renewed by limbal stem cells, located in the limbus and in palisades of Vogt. Palisades of Vogt maintain the clearness of the corneal epithelium by blocking the growth of conjunctival epithelium and the invasion of blood vessels over the cornea. The limbal region can be damaged by chemical burns, physical damage (e.g., by contact lenses), congenital disease, chronic inflammation, or limbal surgeries. The degree of limbus damage is associated with the degree of limbal stem cells deficiency (partial or total). For a long time, the only treatment to restore vision was grafting part of the healthy cornea from the other eye of the patient or by transplanting a cornea from cadavers. The regenerative medicine and stem cell therapies have been applied to restore normal vision using different methodologies. The source of stem cells varies from embryonic stem cells, mesenchymal stem cells, to induced pluripotent stem cells. This review focuses on the use of oral mucosa epithelial stem cells and their use in engineering cell sheets to treat limbal stem cell deficient patients.

## 1. Introduction

A healthy cornea is essential for proper vision. This part of the eye must be kept clear to be fully functional. The cornea is divided into different parts: (1) the corneal epithelium, (2) the Bowman membrane, (3) the stroma, (4) the Descemet membrane, and (5) the corneal endothelium (Scheme 1). The Bowman membrane is located between the corneal epithelium and the stroma and is present throughout the entire mammalian kingdom. The Bowman membrane is innervated randomly over the membrane [1]. The Bowman membrane function is unknown, but it does not appear to be vital for eye function because the absence of a Bowman membrane has no negative impact on patient eyes after undergoing excimer laser photorefractive keratectomy [2]. The stroma creates 80% of the cornea thickness and is composed mainly of water and proteins. Xuan et al. [3] identified 1679 different types of proteins present in the stroma. Keratinocytes were detected in the stroma, which are involved in the secretion of extracellular matrix, which is essential for maintaining the transparency of the cornea [4]. The corneal endothelium is composed of a monolayer of endothelial cells, but its function is crucial. Nutrients from the anterior eye chamber diffuse to the stroma to supply the stroma with nutrients. If excess nutrients diffuse into the stroma, the cornea might swell, and its transparency might decrease. The endothelial cells control the pressure in the cornea by expressing Na^+^/K^+^ pumps, which regulate the osmotic pressure between the stroma and the anterior eye chamber [5]. The corneal epithelium is a first barrier of defense from any external incursion. It is composed of a few layers of epithelial cells tightly connected by complex proteins (tight junction, desmosomes, and hemidesmosomes). These connections and the precise cell organization maintain the transparency of the cornea. The corneal epithelium is constantly renewed by limbal stem cells, located in the limbus. The corneal epithelial cells completely renew in five to seven days [6]. The asymmetric division of the limbal stem cells generates a limbal stem daughter cell and a transient amplifying cell, which migrate to the central cornea. Limbal stem cells migrate toward the middle of the corneal epithelium, in an X, Y, Z direction [7]. During their migration, limbal stem cells differentiate until they become squamous cells and detach from the surface of the cornea [8].

The stem cell niche is precisely located at the level of palisades of Vogt, in the limbus [9]. Injury to the limbal niche prevents corneal epithelial cell renewal and results in the growth of conjunctival epithelium over the cornea. Conjunctival growth over the cornea is accompanied by the neovascularization of the cornea. Many studies reported that the limbal region functions as a barrier between the cornea and conjunctiva. The role of the barrier is to block the conjunctivalization and the neovascularization of the cornea [10,11,12,13]. Damage to this barrier leads to the development of limbal stem cell deficiency (LSCD). The process of conjunctivalization, where conjunctival epithelial cells invade and populate the corneal surface, results in neovascularization, opacification, and inflammatory cell infiltration [14,15,16]. This limbal stem cell deficiency leads to different levels of visual impairment, as reported by the international Limbal Stem Cell Deficiency Working Group [15,17,18]. LSCD can be caused by exogenous trauma, such as thermal burns, chemical injury (alkali burn), or endogenous eye diseases (e.g., Stevens–Johnson syndrome, ocular pemphigoid, aniridia (a genetic disorder), contact lenses, multiple surgeries, or microbial infection) [17,19].

Different methodologies have been developed since 1905, when the first corneal transplantation was completed by Dr. Eduard Zirm [20]. Corneal transplantation is used to repair only the damaged central part of the cornea, but cannot restore the presence of limbal stem cells, which are involved in the renewal of the corneal epithelium, or be used for long term treatment of corneal epithelial defects due to limbal stem cell deficiency. Limbal stem cell transplantation is used to restore the renewal process of the corneal epithelium when the limbal stem cells are damaged and can no longer perform their duty. The curing potency of the limbal stem cell graft can be superior to the corneal transplantation. Grafting of limbal stem cells renews the central part of the damaged cornea to treat corneal epithelial defects [21].

No approved treatment currently exists for bilateral LSCD patients other than: (1) autologous grafting of limbal stem cells [22,23] and (2) allografting of limbal epithelium from a deceased donor [24,25]. Autologous grafts produce excellent results in treating the LSCD cornea because the risk of graft rejection from the transplant is reduced. However, this treatment has limitations: (1) this approach cannot be performed if the patient has bilateral LSCD, which is a challenging task for ophthalmologists [26]; (2) a risk exists of damaging the healthy cornea [14]. Donor cornea allograft treatment heavily depends on the supply of donor corneas provided by eye banks. The shortage of donor eyes is well known worldwide as a serious problem [27]. Even when a donor cornea is grafted onto a patient’s eye, a long term immunosuppressant treatment to decrease the graft rejection risk is required [28,29]. Appendix A summarizes the different types of cells used for corneal regeneration.

## 2. Tissue Engineering Techniques

### 2.1. Autologous Graft and Allograft to Treat LSCD

The first treatment developed to reverse LSCD was transplantation. For patients with unilateral limbal stem cell deficiency, three different methodologies are available. The first one consists of transferring a large limbal graft from the healthy eye to the LSCD eye. This methodology decreases the chances of rejection due to an immunoreaction [30]. However, the risk always exists of triggering an LSCD phenotype on the healthy eye, which can lead to a bilateral LSCD [31]. This approach is not applicable for patients with bilateral LSCD [32]. Even if the limbal graft is the best method to reverse the LSCD phenotype, patients with unilateral LSCD can also be treated with conjunctival limbal autograft [21]. The addition of the conjunctival tissue decreases the size of the limbal tissue surgically removed from the healthy eye, thereby decreasing the risk of LSCD formation on the healthy cornea. However, the addition of the conjunctival tissue increases the risk of conjunctival tissue invasion over the cornea because the barrier function of the limbus was not restored. The second methodology involves using organs from deceased people, which are the easiest to harvest for transplantation. The same approach was used for corneal transplantation. Dr. Zirm performed the first corneal transplantation with some success in 1905 [20]. He transplanted corneas from a deceased boy to the cornea of an adult patient. One of the grafts experienced complications, but the graft on the second cornea maintained the transparency of the patient cornea. Allograft treatment is usually performed for bilateral LSCD by transplanting the cornea of deceased people, and autografts are usually performed for the unilateral LSCD [21,33,34,35,36].

In 1993, Langer and Vacanti introduced a new scientific term: tissue engineering [37]. Tissue engineering is a field combining the fields of biology and engineering to develop artificial tissue able to restore or improve the tissue function. This technology requires the use of stem cells, which can be isolated from any human organ.

### 2.2. LSCD Treatment with Stem Cells

Embryonic stem cells, mesenchymal stem cells, umbilical stem cells, and others have the potential to differentiate into any type of cell, such as myocytes and neurons and epithelial, adipocyte, and epithelial cells [38,39,40,41,42,43,44,45,46,47]. In 2006, induced pluripotent stem cells (iPSCs) were produced by Takahashi and Yamanaka by transfecting stem cells factors (Oct3/4, Sox2, c-Myc, and Klf4) in fibroblasts. These artificial stem cells have the potential to differentiate into all types of cells under adequate cell culture conditions [48].

Progenitor stem cells are another type of stem cell that can be isolated in the organs, but these cells are already specialized to a type of cell, so their capacity to differentiate is limited. For example, cardiovascular progenitor cells can be isolated from the heart [49]. During heart injury, factors are released to activate the proliferation and the differentiation of the cardiovascular progenitor cells to repair the tissue [50,51]. These cardiovascular progenitor cells do not have the capacity to differentiate into other types of cells such as hepatocytes and neurons. However, skeletal myoblasts were studied for repairing heart failure [52]. Oral mucosa epithelial cells (OMECs) can differentiate into different types of epithelium. Due to their biological similarities, oral mucosal epithelial cells were isolated to engineer cell sheets for esophagus treatment after endoscopic submucosal dissection [53,54]. OMECs were also used to engineer cell sheets to reverse limbal stem cell deficiency [55].

The proof of concept was developed by Gipson et al. [56] in 1986 when an oral mucosal epithelium biopsy was transplanted to damaged rabbit cornea. This study initiated the development of the use of oral mucosal epithelial cells to repair damaged cornea. Only in 2003 was a second study performed on a rabbit model by transplanting oral mucosal epithelial cell sheets on keratectomized rabbit eyes grown on amniotic membrane [57]. In 2004, the first human study was conducted. Nishida et al. [55] recruited four patients afflicted with limbal stem cell deficiency. In this study, the authors used a thermoresponsive surface to grow and harvest the cell sheets without any extra scaffold or carrier, as used in other studies [58,59]. Reviews were written about the treatment of limbal stem cell deficiency using progenitor stem cells and cell sheets [60,61,62], mentioning the strong potential of this treatment for LSCD.

We summarize the oral mucosa epithelial cells sheets used to reverse the limbal stem cell deficiency in clinical trials.

### 2.3. Treatment of Limbal Stem Cells with Oral Mucosa Epithelial Cell Sheets

Among the cells used for corneal recovery, oral mucosa epithelial cells are the most commonly used for in vitro, in vivo, and translational applications. Nakamura and Kinoshita engineered a stratified cell sheet by culturing oral mucosa epithelial cells (OMECs) on amniotic membranes. The cell sheets were grafted on keratectomized rabbit eyes and successfully recovered vision with a clear cornea [57]. Cell sheets were found to be positive for keratin 3 (*KRT3*) and keratin 13 (*KRT13*), which are expressed naturally on the central cornea, and keratin 4 (*KRT4*), which is absent on the central cornea, but present on the oral mucosa epithelium and the conjunctiva. Cell sheets engineered on amniotic membranes were analyzed in depth to validate the different parameters of the cell sheets: differentiation, cell–cell contact, and renewal potential of the cells [63]. Epithelial stem cell markers *KRT15*, δ*Np63*, and *p75* were identified on the cell sheet. The presence of *δNp63* indicates a long term cell sheet capacity renewal because its knock-out impairs epithelial growth [64]. Connexin 43, a gap junction protein, was also detected, which is involved in cell–cell communication and in the maintenance of strong cell–cell contact to limit the cell migration risk. Finally, markers of transdifferentiation were also detected, such as *KRT3*, *KRT4*, and *KRT13*. Other laboratories used OMECs to engineer stratified cell sheets using a similar protocol. For example, Nishida et al. [55] cultured OMECs on a temperature sensitive cell surface in the presence of NIH 3T3 feeder cells (3T3 means “3-day transfer, inoculum 3 × 10^5^ cells”). An amniotic membrane was not used as a carrier, but instead, oral mucosal epithelial cell sheets were harvested by decreasing the ambient temperature to allow the cell sheet to detach from a thermosensitive surface [55]. In both studies, mouse feeder cells were used to assist with the multilayer cell sheet engineering, which was reported by most studies. In translational applications, OMEC sheets showed promising results for repairing damaged corneas (Stevens–Johnson’s syndrome and chemical and thermal injuries) [65,66,67,68,69]. OMEC sheets were reported to have higher invasion and migration properties than corneal epithelial cells. Furthermore, OMEC sheets secreted more fibroblast growth factors 2 (FGF2) (but not vascular endothelial growth factor (VEGF)) than corneal epithelial cells [70]. Because the success rate of transplanted OMEC sheet is not 100%, the results reported about migration and FGF2 expression explain the patient failures. However, the FGF2, migration and invasion study was only performed in vitro on human umbilical vein endothelial cells [70].

A different study concluded that the expression of FGF2 is similar between corneas and OMEC sheets after grafting. This model is closer to reality and indicates that OMEC sheets do not have angiogenic potential [71]. Notably, isolated OMECs from each patient are different, with different phenotypes due to the donor’s health conditions, lifestyle, and genetics; these factors can explain the success or failure of transplanted OMEC sheets.

### 2.4. Carrier for Cell Sheet Grafting

Engineering cell sheets with Food and Drug Administration (FDA) approved products is a difficult task, but their harvesting requires specific protocols and growing surfaces. Different methodologies were developed to facilitate corneal-like cell sheet harvesting: fibrin glue [72], contact lenses [73], amniotic membranes [74], thermo-responsive surfaces [55], and dispase [65,75]. The last three methodologies were used for human studies.

For the majority of the reported clinical trials, amniotic membranes were used to harvest and graft the cell sheet (71.4% of the clinical trials). Amniotic membranes were, at first, directly grafted onto the cornea, before being used as a cell sheet carrier [76]. Tseng et al. [77] showed that the addition of allograft limbal stem cells to the amniotic membrane improved the healing process and the corneal epithelium recovery when both were grafted on the cornea. Similar results were obtained when the corneal epithelial cells were cultured on amniotic membranes [78]. For total epithelium defects, the combination of epithelial cells with an amniotic membrane produced higher and faster reepithelization of the cornea compared to the graft with only the amniotic membrane [77,79]. In both of these cases, amniotic membranes were used as a carrier for cell sheet grafting and as feeder cells. Amniotic membranes are widely used for cornea cell sheet transplantation [79,80,81]; however, the availability is limited, and the membranes require the use of sutures to attach the cell sheet to the cornea. Mouse fibroblast feeder cells are used in co-culture with the amniotic membrane [59,67,68,69,82,83,84,85,86,87,88,89].

An innovative approach involved the use of a thermoresponsive surface that detaches the cell sheet from the culture by decreasing the temperature to below 32 °C. PolyN-isopropylacrylamide was developed in 1968 by Heskins and Guillet [90]. The hydrophobicity of this intelligent polymer varies based on temperature. By becoming hydrophilic below 32 °C, the water penetrates under the cell sheet and detaches it without damaging the cell sheet or the extracellular matrix at its bottom. Numerous types of cells were successfully tested using a thermoresponsive surface: keratinocytes, endothelial cells, hepatocytes, cardiomyocytes, oral mucosa epithelium cells, and corneal epithelial cells [91,92,93,94,95]. Nishida and colleagues pioneered the use of the thermoresponsive surface to manufacture cell sheets for limbal stem cell deficiency. In two studies (9.52% of the studies), stratified cell sheets were engineered on an intelligent surface in the presence of mouse feeder cells and grafted on limbal stem cell deficient eyes [55,66]. The polymer did not affect the stratification of the cell sheet, the polarity of the cell sheets, or the stemness of the seeded progenitor stem cells. After harvesting, cell–cell junctions and the extracellular matrix were maintained after the cell sheet harvesting.

A gentle enzymatic approach was used by Kim in two human studies [65,75] (9.52% of the human studies). Engineered cell sheets were detached using dispase treatment, which cleaves collagen and fibronectin from the extracellular matrix [96]. In these studies, cell sheet morphology and phenotype were not studied, but the results after transplantation showed that the cell sheet curative properties were not altered by dispase treatment.

All these approaches did not require sutures to maintain the cell sheet on the cornea, which is an important feature for healing damaged corneas. However, the majority of the clinical trials reported the use of sutures (usually 10-0) to maintain the cell sheet on the top of the cornea, with the use of contact lenses to protect the cell sheet from mechanical damage from the eyelid.

## 3. Steps Prior to Cell Sheet Graft

Before the transplantation of the cell sheet to the cornea, different steps and data must be recorded. Appendix A summarizes the information recorded about the patients, the cell sheet, the protocol to engineer the cell sheet, etc. For all studies, the inclusion criteria were the total loss of palisade of Vogt, total LSCD, and minimal visual acuity. Some studies did not report the exclusion and inclusion criteria. The age of the patients, unilateral or bilateral LSCD, or the level of inflammation of the cornea were not used as exclusion or inclusion criteria.

Exclusion criteria were not mentioned in the publications. We hypothesize that exclusion criteria could be related to oral infection or any infection (such as hepatitis B virus, hepatitis C virus, human immunodeficiency virus, and syphilis), neoplastic diseases, pregnancy, congenital diseases (e.g., aniridia), or if the patients are unable to fulfil the minimal requirements of the study.

### 3.1. Treated Patients

The etiology of LSCD is variable. In total, 249 patients were recruited in clinical trials of LSCD, caused by: Stevens–Johnson syndrome (*n* = 89), chemical burn (*n* = 114), ocular cicatricial pemphigoid or pseudo-ocular cicatricial pemphigoid (*n* = 57), and thermal burn (*n* = 6). Among them, the factors related to LSCD corneas were variable and could play an important role in the outcome of the transplantation grafting (not transplantation).

The average age of the patients was 50.6 years old, ranging from eight to 86. The donor’s age might play a role in the outcome of the LSCD treatment. No data were reported about the potential relationship between patient age (the age of the oral mucosal epithelial cells) and the outcome of the treatment after grafting. A review reported that in vivo aging mesenchymal stem cells showed decreased proliferation potential, differentiation potential, and telomerase length and an increase in genetic instability [97]. The number of mesenchymal stem cells isolated from tissues can decrease with age [98], which could also be the case for oral mucosal epithelial cell donors. However, no information has been published about the number of cells isolated per patient or about their potency. The decrease in MSC proliferation and differentiation potential was reported with the reduction in the colony forming assay [98,99]. Aged people were reported to have a greater number of larger and flatter cells in the oral mucosal epithelium, which could be related to a decrease in progenitor cells [100]. In addition to their decrease in cell proliferation, it was also reported that mesenchymal stem cells are more sensitive to oxidative stress, undergo more apoptosis, and their capacity to differentiate is decreased [98,99,101]. Different protocols were suggested as solutions to improve cell viability and resistance to aging, such as the use of xeno-free expansion culture media [102] or drugs [103,104]. Data obtained from aging MSC could be used to improve the treatment of LSCD with oral mucosa epithelial cells (e.g., composition of the culture media).

The definition of the disease is one of the criteria that is important to report to understand the outcome of the treatment. We did not review the definition of the LSCD; a complete review was published by The International Limbal Stem Cell Deficiency Working Group [15]. Only total LSCD, with the loss of palisades of Vogt, were recruited in these studies. If the central cornea was transparent, the patients were not recruited. For most studies, Schirmer’s test was not performed on the patient, especially for severely dry eyes. Based on Shimazaki’s study, the outcome of transplantation could be improved if the Schirmer’s test is above 10 mm [105], but only 14.3% of the studies reported the use of the Schirmer’s test. For the inflammation status, the inflammation must be under control or it can jeopardize the success of treatment [106]. Usually, inflammation occurs after corneal surgery, which is treated with the injection of steroids [86]. Additional surgeries on the eye must be performed due to abnormalities such as symblepharon or fornix [66,68,86]. Many patients have these abnormalities, and they should be treated before the cell sheet transplantation is carried out to improve the treatment outcome [66]. In some studies, symblepharon or abnormalities were reported, and these were related to the patient outcome. All these factors (abnormalities, inflammation, and duration of LSCD) should be recorded and connected per patient to determine a potential relationship between a successful or failed transplantation.

Of the studies, 85% used NIH 3T3 fibroblasts as feeder cells and 71.4% used amniotic membranes. Fetal bovine serum (FBS) or autologous serum was used to grow the epithelial cells into cell sheets in 47.2% of the studies. Bovine serum or derived bovine products are not fully rejected by the FDA, but the manufacturers must provide the origin of the products [107]). In some studies, the authors used autologous serum for the growth of the cell sheets [67,84,87,108] or the authors reported the use of bovine fetal serum and autologous serum without mentioning how the cell sheets were engineered with the serum [59,69,82,109]. For FBS, each autologous serum lot was variable between patients, adding a variable that could affect the cell sheet engineering manufacturing process, but allowed the patient to use their own serum with their own cells. However, FDA recommends the use of chemically well defined culture media for translational purposes in the absence of animal products [110,111] (FDA source Class II Special Controls Guidance Document: Tissue Culture Media for Human ex vivo Tissue and Cell Culture Processing Applications; Final Guidance for Industry and FDA Reviewers). This was the case in 19% of the reported studies for at least the serum used in cell culture (Appendix A).

### 3.2. Characterization of Biopsy and Cell Sheet

None of the studies reported the number of oral mucosa epithelial cells isolated from a biopsy. This number is critical information when engineering several cell sheets. The surface of the biopsy ranges from 2 [57] to 200 mm^2^ [65], and we speculate that it is possible to isolate 100 times more cells with the 200 mm^2^ biopsy. Because problems can occur during the transplantation procedure (delay and problems during the transport of the cell sheets, harvesting problems, and cell sheets breaking during the harvesting and during the cell sheet grafting surgery), several cell sheets should be engineered from the same biopsy. At least two cell sheets should be shipped to the surgeon, and at least one cell sheet should be studied in the laboratory for morphology and phenotype determination. Extra cell sheets could even be cryopreserved for long term storage if only a small portion of the cell sheet is required [112].

The quality of the biopsy is another important factor that can affect the treatment outcome because the raw material for engineering the cell sheets is isolated from biopsies from patients with a wide age range (eight to 86 years old), and with different health issues, including smokers, alcohol consumers, and diabetic patients [113,114,115,116,117]. Among all the clinical trials, few studies reported biopsy [68] or patient health data, which is understandable because the size of the biopsy is very small, but aging might affect cell biopsy [100]. Inatomi, with a 3–5 mm^2^ biopsy, was able to identify the morphology and the expression of *KRT3* and the absence of expression of *KRT12*. Satake reported the morphology using hematoxylin and eosin (H&E) staining of the oral mucosa epithelial cells [67]. Priya identified the expression of p63 and the morphology of the oral mucosa epithelium [87]. The oral mucosal epithelial cell sheets have a similar morphology (stratified multilayer cell sheets) to the oral mucosa epithelium and the corneal epithelium. The progenitor epithelial cells are present on the basal side, with squamous cells on the apical side. For all these studies, the authors reported the data from one oral mucosa from one patient per publication, while the studies included two patients in Inatomi, four in Satake, and ten in Priya. In total, 1.2% of the oral mucosa biopsies were characterized by immunostaining or morphology.

The characterization of the oral mucosa is an important criterion in terms of engineering the cell sheet and the outcome of its grafting on LSCD cornea. *p63* expression was analyzed in only one biopsy [87]. *p63* is known to play a major role in the renewal of epithelium and cornea [118]. The knockout of *p63*, in a mouse model, was related to the decrease in epithelium renewal and to final differentiation of the epithelial cells. Inflammation of oral mucosa can decrease the expression of *p63*, an important transcription factor involved in the proliferation of epithelial cells [118]. In patients with oral lichen planus and graft vs. host disease, the expression of all *p63* isoforms decreased with oral inflammation [119]. Based on this reported information, the time of biopsy collection must be planned once the inflammation is well managed on the oral mucosa. Some reports indicated that the expression of p63 isoforms could change with aging. *δNp63* is known to have an anti-apoptotic effect, and TAp63 is a pro-apoptotic protein [120]. The expression of *TAp63* decreases in the lungs of aging mice [121]. The specific functions of *p63* isoforms are still unclear and require additional investigation. p63 is known to play an important role in epithelium renewal. Determining the level of *p63* expression could provide information that would affect the production of cell sheets and the treatment outcome. For example, Rama et al. [33] suggested that a certain percentage of cells expressing p63 in the cell sheet ensures treatment success. In Rama et al. [33], cell sheets were engineered with limbal stem cells and not oral mucosa epithelial cells. However, this information could be part of the cell sheets release criteria.

Cell sheet characterization is important before transplantation, as part of the release criteria, confirmed using immunohistochemistry (IHC) or quantitative real-time PCR. Only some of the clinical trials reported the morphology of the cell sheets, and only 33.3% reported the expression of different proteins (*p63, ABCG2, p75, PAX6, β1-Integrin, laminin5, KRT3,* and *KRT12*) before transplantation. In most studies, no information about the biopsy or the cell sheets was published. The expression of important factors (i.e., stem cell markers, cell–cell connection) should be determined from a small part of the cell sheets as was performed for the oral mucosa biopsy [67,68] including: morphology, balance of the angiogenic factors, cell–cell connection, cell proliferation, and differentiation of progenitor cells into squamous cells.

## 4. Transplantation and Post-Surgery

The surgical procedures were similar in all the studies. To treat eye abnormalities such as symblepharon and conjunctival fornix, the conjunctiva area around the cornea was treated with 0.04% of mitomycin C to block the growth of conjunctival cells [67,68,84,89]. For cell sheet grafting, a 360 degree conjunctival peritomy was performed up to 3 mm outside in the conjunctiva to expose the cornea. After grafting, the cell sheets were grafted with sutures in some clinical trials [59,68,69,82,85,87,88] and not in others [55,66]. A soft contact lens was placed on the top of the cornea to protect the cell sheet in certain studies [65,66,83,85,87,88,109].

The follow-up, adverse events, and severe adverse events for the patients are reported in Appendix A.

### 4.1. Adverse Events and Severe Adverse Events

For researchers, the priority is the safety and health of patients recruited in clinical trials. Clinical trials reported adverse events (AEs) that affected recruited patients. AEs can directly occur due to the product, via rejection, or can occur due to the surgical procedure, such as infection, corneal perforation, symblepharon formation, etc. A total of 55 AEs was reported, with only nine severe adverse events (SAEs) among the 249 treated patients.

Among the 55 AEs, persistent epithelial defects (PEDs) were the most frequent (30 cases), and they were recurrent after the initial cell sheet transplantation (54.54% of the total AEs). A review explained how persistent epithelial defects are treated [122]. In some of the clinical studies, the PEDs resolved by themselves with the help of grafted cell sheets or conjunctival cells [69,82,123]; antibiotics were used [86], a second reoperation was performed (oral mucosa epithelial cell sheet, amniotic membrane, allogenic corneal, or limbal stem cells transplants) [65,69,123], with a contact lens bandage [89,108]; or they did not resolve [59,66,67,109,124].

Intraocular pressure was reported for 14 of the patients (5.62% of all patients), which resolved with specific treatment, such as antiglaucoma medication or carbonic anhydrase inhibitor [88,124].

Infection of the cornea was detected in four patients (1.6% of the total number of patients), and the infections resolved with antibiotic treatment [59,86,123,124]. Some other AEs were recorded, such as cataract, pain, corneal recurrence, Meibomian cyst, keratitis, symblepharon formation, drug induced allergy, and liver dysfunction, which resolved after stopping a systemic drug treatment [59,66,83].

Nine SAEs were reported: two corneal perforations, where one cornea healed with a small patch graft [85], and the patient in the other study was withdrawn [66]; seven cell sheets were rejected [65,66,83,123].

### 4.2. Follow-Up and Visual Acuity

To minimize the drug treatment of the patient as drugs have a short term effect, the goal of cell therapy is to reverse a disease for life. The average time for the follow-up of the patients was around two years, with a maximum of 7.5 years, for all the clinical trials involving oral mucosal epithelial cell sheets to treat LSCD. Based on the FDA Guideline for Long Term Follow-Up After Administration of Human Gene Therapy Products updated in July 2018, the studies should annually follow-up with the patients for five years, with an extra 10 years of annual queries to ensure the long term safety of the patients. The FDA does not provide clear guidelines for cell therapy (such as for cell sheet technology), but the FDA requires more than a one year follow-up for the patients (Considerations for the Design of Early-Phase Clinical Trials of Cellular and Gene Therapy Product). The one year follow-up was used by all the reported clinical studies discussed here; however, for 21 patients (8.4%), the follow-up was less than one year [69,82,88]. Five years with annual follow-ups was applied by only two clinical trials for LSCD treated with an oral mucosal epithelial cell sheet for some of their patients [59,124].

The visual acuity improved for the majority of the patients treated with oral mucosal epithelial cell sheets. For 246 transplanted corneas, 52.8% of the corneas demonstrated an improvement, 8.2% were steady, and 2.9% deteriorated. Based on a review, the transplant of limbal stem cells was over 70% successful, and it is thought that this success rate can be increased. We think that oral mucosal epithelial cell sheet technology to treat LSCD can also be improved, because only 52.8% of the transplanted cornea resulted in improved visual acuity.

Neovascularization of the grafted cornea occurred [55,86,108], but never grew over the cornea, indicating that cultured autologous oral mucosal epithelial cell sheet (CAOMECS) can block the neovascularization of the cornea of the healthy cornea [125]. The mechanism of action of the cell sheet is not well understood, but it could involve a combination of the physical barrier and the production of anti-angiogenic factors. The physical barrier function is indicated by the decrease in the corneal conjunctivalization level after the epithelium cell sheet graft, but this is not always the case [88,124].

Corneal opacity is another aspect of patient outcome. Corneal opacity decreased over time after a cell sheet graft in some studies [55,86]. If the opacity did not improve and was persistent, penetrating or deep lamella keratoplasty (PKP and DALK, respectively) was performed [59,65,68,83,87,123]; in some cases, two years after the initial grafting [85,88]. PKP and DALK should only be performed after cell sheet graft and once the cornea surface is stable.

## 5. Conclusions

In most cases, cells are grown using animal products such as serum or trypsin. For example, fetal bovine serum, bovine serum, and bovine pituitary extract (BPE) are usually introduced into the manufacturing process to ensure the growth of the cells during the culture period [65,86,87,126,127]. After the Creutzfeldt–Jakob epidemic, BPE was prohibited in cell manufacturing for cell therapy. In the United Kingdom, around 2000 people were contaminated with the prion, the cause of Creutzfeldt–Jakob disease [128]. The factor causing Creutzfeldt–Jakob disease may be present in BPE and may be able to be transmitted to the patient after cell therapy. Animal sera can be imported from New Zealand, and their cleanness is certified. However, the FDA advises not to use them if replaceable products are available for developing a chemically defined culture media. Engineering a stratified multilayer epithelial cell sheet using a chemically defined culture media that is not yet available on the market is a challenging task. The final cell sheet product must have the desired epithelium-like phenotype to maintain the renewal capacity of the cells. Other characteristics are essential (stemness, cell–cell connection, presence of anti-angiogenic factors), as described in the corneal epithelium to decrease the risk of ectopic migration of grafted cells and to block or decrease the neovascularization over the cornea [70,129,130]. The characteristics of the cell sheet should be better understood before the transplant using a small cut of the cell sheets.

In summary, the methodologies and the protocols used to treat patients afflicted with limbal stem cell deficiency should be improved by increasing collaboration and communication. Standardization of oral mucosal epithelial cell sheet engineering and characterization of the cell sheet will help with the comparison of the patient outcomes. Scheme 1 depicts the cornea anatomy and the cellular models used for the corneal regeneration on limbal stem cell deficient corneas. Scheme 2 summarizes the criteria that are important for the epithelium cell sheet engineering.

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
