# Peer review of "Clinical Trials of Limbal Stem Cell Deficiency Treated with Oral Mucosal Epithelial Cells"

_ijms, 2020, doi:10.3390/ijms21020411_

Round 1
Reviewer 1 Report
The review of Oliva et al. discusses the use of oral mucosal epithelial cells for the treatment of limbal stem cell deficiency resulting in severe loss of vision. The topic is of interest to the readers of the IJMS but the manuscript is poorly organized and the language needs substantial revision.
Following introduction, the description of the technology used and of clinical applications could be reorganized as follows.
Tissue engineering techniques Clinical trials: describe type (I, II, etc) and inclusion criteria Clinical outcome, adverse events …
Additional observations
Page 1, Introduction:
A figure would help understanding the anatomy, here described in detail
Discuss corneal transplantation vs. limbal stem cell transplantation, i.e. how the latter could overcome the limitations of the former.
Page 2, Autologous graft and allograft to treat LSCD:
some fundamental references are missing (e.g. NEJM!!)
Page 4, Treated patients
The discussion on aging MSCs is misleading, since these cells are not used to treat LSCD.
Page 8, lines 43-45. This statement is incorrect. According to FDA, minimal manipulation is not the only requirement for allowing direct reinjection of autologous cells. These cells should also be intended for homologous use, i.e. “the repair, reconstruction, replacement, or supplementation of a recipient's cells or tissues with an HCT/P that performs the same basic function or functions in the recipient as in the donor”. Thus, the use of oral epithelial cells is NOT homologous.
Page 8, Conclusion. The Conclusion is a continuation of the discussion, while it should be limited to a few “take home” messages.
Author Response
The review of Oliva et al. discusses the use of oral mucosal epithelial cells for the treatment of limbal stem cell deficiency resulting in severe loss of vision. The topic is of interest to the readers of the IJMS but the manuscript is poorly organized and the language needs substantial revision.
Following introduction, the description of the technology used and of clinical applications could be reorganized as follows.
Tissue engineering techniques Clinical trials: describe type (I, II, etc) and inclusion criteria Clinical outcome, adverse events …
Author's answer:
Thank you for the reviewer’s comment. We reorganized the manuscript following the suggestions of the reviewer. We also included the inclusion and exclusion criteria in the paragraph 3.
Additional observations
Page 1, Introduction:
A figure would help understanding the anatomy, here described in detail
Discuss corneal transplantation vs. limbal stem cell transplantation, i.e. how the latter could overcome the limitations of the former.
Author's answer:
We added a figure to show the anatomy of the cornea (scheme 1), that we described in the introduction. In the introduction, we added the discussion and the comparison of the corneal transplantation vs. limbal stem cell transplantation, as recommended by the reviewer.
Page 2, Autologous graft and allograft to treat LSCD:
some fundamental references are missing (e.g. NEJM!!)
Author's answer:
We added references in the paragraph, reporting the autologous graft and the allograft graft to treat LSCD.
Page 4, Treated patients
The discussion on aging MSCs is misleading, since these cells are not used to treat LSCD.
Author's answer:
We agree with the comment of the reviewer. MSC are not used to treat LSCD but our point was to use our knowledge about the MSC age influencing the outcome of the treatment. There is no study reporting a potential influence of the oral mucosal epithelial cells age on the outcome, after the graft on the cornea besides the study conducted by Abu Eid, which is reported in the same paragraph.
Page 8, lines 43-45. This statement is incorrect. According to FDA, minimal manipulation is not the only requirement for allowing direct reinjection of autologous cells. These cells should also be intended for homologous use, i.e. “the repair, reconstruction, replacement, or supplementation of a recipient's cells or tissues with an HCT/P that performs the same basic function or functions in the recipient as in the donor”. Thus, the use of oral epithelial cells is NOT homologous.
Author's answer:
Thank you for the comment. We removed the part mentioning the minimal manipulation of the cells from the paragraph.
Page 8, Conclusion. The Conclusion is a continuation of the discussion, while it should be limited to a few “take home” messages.
Author's answer:
We agree with the reviewer comment. We corrected the discussion to make it shorter, with few “take home” messages.
Reviewer 2 Report
The review article on "Clinical Trials of Limbal Stem Cell Deficiency Treated with Oral Mucosal Epithelila Cell Sheets" is very well written. This manuscript is unique in the literature it cover, It start with very basic research finding through a very advance clinical application, which make a unique from currently available literature. The available review articles on cornea regeneration are based on one particular cell type or technique, while this manuscript has presented unique literature from very basic to advanced clinical studies covering a range of cell types. The present manuscript has a unique feature of comparing different cell types used for cornea regeneration, including cell sheets, stem cells, mucosa cells, autologus and allograft cells.
The authors organized the literature in very well. All the available options for the regeneration of cornea are explored in details.
This review is very comprehensive on the topic, and must be published with priority.
Minor Revision:
The title of the manuscript need to be modified, its about cell sheets, while in the content a number of cellular based approaches for corneal regeneration were discussed.
A cartoon figure need to be introduced to highlight the cellular models used for corneal regeneration, although a scheme 1 is there but still a cartoon image or figure addition is required.
A table can be introduced in which different types of cell, their properties and cornea regeneration potential, and reference for clinical studies can be highlighted.
Author Response
Comments and Suggestions for Authors
The review article on "Clinical Trials of Limbal Stem Cell Deficiency Treated with Oral Mucosal Epithelila Cell Sheets" is very well written. This manuscript is unique in the literature it cover, It start with very basic research finding through a very advance clinical application, which make a unique from currently available literature. The available review articles on cornea regeneration are based on one particular cell type or technique, while this manuscript has presented unique literature from very basic to advanced clinical studies covering a range of cell types. The present manuscript has a unique feature of comparing different cell types used for cornea regeneration, including cell sheets, stem cells, mucosa cells, autologus and allograft cells.
The authors organized the literature in very well. All the available options for the regeneration of cornea are explored in details.
This review is very comprehensive on the topic, and must be published with priority.
Minor Revision:
The title of the manuscript need to be modified, its about cell sheets, while in the content a number of cellular based approaches for corneal regeneration were discussed.
Author's answer:
We would like to thank the reviewer for the comment. The title was modified.
A cartoon figure need to be introduced to highlight the cellular models used for corneal regeneration, although a scheme 1 is there but still a cartoon image or figure addition is required.
Author's answer:
We drew a cartoon explaining the development of the LSCD and the treatment of the LSCD by the cell sheet grafting on the cornea. In addition, a cartoon explaining the morphology of the cornea was included (Scheme 1)
A table can be introduced in which different types of cell, their properties and cornea regeneration potential, and reference for clinical studies can be highlighted.
Author's answer:
Thank you for the comment. We made a table (Table S1), that summarizes the different type of cells used to regenerate the cornea, in clinical trials.
Round 2
Reviewer 1 Report
The Authors have answered satisfactorily to all my comments.